Refining the reaction mechanism of O2 towards its co-substrate in cofactor-free dioxygenases

Silva Pedro J. pedros@ufp.edu.pt
FP-ENAS/Fac. de Ciências da Saúde, Universidade Fernando Pessoa , Porto , Portugal
Schweitzer-Stenner Reinhard
Electronic publication date: 2016 Dec 20
Publication date: 2016
Volume: 4
Electronic Location ID: e2805
Received 2016 Jun 20; Accepted 2016 Nov 19
Copyright: ©2016 Silva
Copyright year: 2016
Copyright holder: Silva
License: This is an open access article distributed under the terms of the Creative Commons Attribution License, which permits unrestricted use, distribution, reproduction and adaptation in any medium and for any purpose provided that it is properly attributed. For attribution, the original author(s), title, publication source (PeerJ) and either DOI or URL of the article must be cited.
License URL: https://creativecommons.org/licenses/by/4.0/

Keywords: Computational chemistry, Minimum-energy crossing point, Oxygenase, Density-functional theory, Urate oxidase, Glutamate decarboxylase, Ring cleaving dioxygenase, DFT

Funding: FEDER through Programa Operacional Factores de Competitividade–COMPETE FCT–Fundação para a Ciência e a Tecnologia UID/Multi/04546/2013 The author received no funding for this work, which was performed using computational resources acquired under a previous project (PTDC/QUI-QUI/111288/2009) financed by FEDER through Programa Operacional Factores de Competitividade–COMPETE and by Portuguese Funds through FCT–Fundação para a Ciência e a Tecnologia. The FP-ENAS Research Unit further receives some support from additional Portuguese Funds through a grant from FCT–Fundação para a Ciência e a Tecnologia (UID/Multi/04546/2013). The funders had no role in study design, data collection and analysis, decision to publish, or preparation of the manuscript.

==============================
Cofactor-less oxygenases perform challenging catalytic reactions between singlet co-substrates and triplet oxygen, in spite of apparently violating the spin-conservation rule. In 1-H-3-hydroxy-4-oxoquinaldine-2,4-dioxygenase, the active site has been suggested by quantum chemical computations to fine tune triplet oxygen reactivity, allowing it to interact rapidly with its singlet substrate without the need for spin inversion, and in urate oxidase the reaction is thought to proceed through electron transfer from the deprotonated substrate to an aminoacid sidechain, which then feeds the electron to the oxygen molecule. In this work, we perform additional quantum chemical computations on these two systems to elucidate several intriguing features unaddressed by previous workers. These computations establish that in both enzymes the reaction proceeds through direct electron transfer from co-substrate to O2 followed by radical recombination, instead of minimum-energy crossing points between singlet and triplet potential energy surfaces without formal electron transfer. The active site does not affect the reactivity of oxygen directly but is crucial for the generation of the deprotonated form of the co-substrates, which have redox potentials far below those of their protonated forms and therefore may transfer electrons to oxygen without sizeable thermodynamic barriers. This mechanism seems to be shared by most cofactor-less oxidases studied so far.

Introduction

Reactions where the number of unpaired electrons changes as reactants are transformed into products are not generally allowed by quantum mechanics due to Wigner’s spin-conservation rule. This rule prevents the dioxygen molecule, which has two unpaired electrons and a S = 1 (triplet) ground state, from easily reacting with acceptors in the singlet state (S = 0, like most organic molecules) to yield organic products without unpaired electrons. Such reactions are not, however, strictly impossible due to the intervention of spin–orbit coupling, which mixes both spin states. Understanding spin–forbidden reactions requires the characterization of the potential energy surfaces of the different spin-states involved in the reaction, and the location of the point where both surfaces touch each other (Harvey, 2007; Harvey, 2014). At this specific geometry (the “minimum-energy crossing point”, henceforth abbreviated as MECP) the system may transition (or “hop”) between spin systems, with a probability which depends on the magnitude of the spin–orbit coupling and may be computed approximately according to the Landau–Zener equation (Nakamura, 1987). Since spin–orbit coupling is a relativistic effect which increases with the nuclear charge, formally spin–forbidden reactions become progressively easier as one descends down the periodic table, to the point that “spin-forbidden” transitions involving Ni, Cu or elements of the 5th (or lower) periods are as probable as “spin-allowed” transitions (Marian, 2001). Proteins which generate, use or transport dioxygen therefore usually rely on transition metals such as manganese, copper and iron (Ferguson-Miller & Babcock, 1996; Que & Ho, 1996; Sono et al., 1996; Wallar & Lipscomb, 1996; Yachandra, Sauer & Klein, 1996). A large class of enzymes devoid of metals (the flavoproteins) circumvents the problem with the help of flavin, which readily transfers one electron to oxygen yielding a separated radical pair consisting of a superoxide anion and a flavin-based radical (Massey, 1994; Mattevi, 2006). This mechanistical proposal has found experimental support from the observation of superoxide leakage to the reaction medium from the reaction of protein-bound flavins with dissolved oxygen (Massey et al., 1969; Boubacar et al., 2007; Daithankar et al., 2012) and spectroscopic identification of the flavin semiquinone radical in glycolate oxidase (Pennati & Gadda, 2011). A fast “flip” of the spin in the superoxide anion facilitated by strong spin–orbit coupling in the superoxide anion (−160 cm−1 (Huber & Herzberg, 2016)) then brings the pair to the singlet state, enabling straightforward, non-spin-forbidden reactivity. The reaction then proceeds through radical recombination (yielding a hydroperoxyflavin with no unpaired electrons, in monooxygenases (Mattevi, 2006; Visitsatthawong et al., 2015)) or transfer of a second electron from the flavin semiquinone to superoxide, yielding hydrogen peroxide (Mattevi, 2006).

Several enzymes catalyze the addition of oxygen to suitably π-conjugated substrates in spite of lacking flavin or metals in their active sites, often through “substrate-assisted catalysis” (Fetzner & Steiner, 2010) which takes advantage of the enhanced reactivity of these conjugated system upon enzyme-promoted substrate deprotonation. Extensive computational details of the reaction mechanisms of coproporphyrinogen oxidase (Silva & Ramos, 2008) and vitamin K-dependent glutamate carboxylase (Silva & Ramos, 2007) confirmed that substrate deprotonation is indeed required for their catalytic action. Evidence for substrate deprotonation is also available for urate oxidase (Bui et al., 2014), although in this instance a more complex mechanism involving transient protein-based free radicals was proposed to be operative, based on EPR measurements of anaerobic preparations of substrate-bound enzyme (Gabison et al., 2011). Based on the reaction profile towards a superoxide-scavenging spin probe, radical-pair reactivity towards O2 has also been suggested to occur (Thierbach et al., 2014) in a bacterial ring-cleaving 2,4-dioxygenase active towards (1H)-3-hydroxy-4-oxoquinolines (EC 1.13.11.47), but recent computational results have been interpreted as contradicting this hypothesis, as the computed reaction energy for the electron transfer from substrate to O2 (8–11 kcal  mol−1) would imply an “endothermic process […] unlikely to happen spontaneously in the protein or in solvent” (Hernández-Ortega et al., 2015). As an alternative, these workers computed the energetics of a pathway (Fig. 1) consisting of addition of triplet oxygen to the deprotonated substrate (yielding a triplet peroxide, 3I1, 17 kcal mol−1 above the reactants), followed by a transition to a singlet peroxide state (1I1, 8 kcal mol−1 below the triplet state, i.e., 9 kcal mol−1 above the reactant state). Minimum-energy crossing points between the singlet and triplet potential energy surfaces were not located in that work, but were predicted to lie around 10 kcal mol−1 above the reactant state.

Figure 1 Proposed mechanism for the reaction catalyzed by 1-H-3-hydroxy-4-oxoquinaldine-2,4-dioxygenase.

Some of the conclusions of the computational work of Hernandéz-Ortega et al. seem problematic: on the one hand, the computed 8–11 kcal mol−1 endothermicity of the electron-transfer process from substrate to triplet oxygen does not seem enough to discard the possibility of an electron-transfer mechanism, since larger activation free energies of 17.4 kcal mol−1 are able, according to transition state theory, of sustaining reaction rates of 1 s−1 at room temperature; and on the other hand, the lack of data on the relative energies of the excited singlet state of the triplet peroxide intermediate and of the triplet state of the singlet peroxide intermediate leave open the possibility that those two potential energy surfaces do not cross between 3I1 and 1I1, and that the minimum-energy crossing point actually lies between the reactant state (3R) and the singlet intermediate 1I1, thereby completely bypassing the putative triplet peroxide (3I1). In this work we perform additional computational studies of the putative intermediates of this reaction and conclude that the triplet peroxide state is never formed: the reaction instead proceeds directly from 3R to 1I1 either through low-lying minimum-energy crossing points or, most likely, through direct electron-transfer from substrate to oxygen, yielding a separated radical pair. Additional computations in model systems for other reactions catalyzed by cofactor-less oxygenases strongly suggest that formation of such radical pairs should be the norm for inductively activated π-conjugated substrates.

Computational Methods

Quantum chemical computations were performed with the Firefly (Granovsky, 2013) quantum chemistry package, which is partially based on the GAMESS (US) (Schmidt et al., 1993) source code. As in the original work (Hernández-Ortega et al., 2015), all computations were performed with the B3LYP density-functional (Lee, Yang & Parr, 1988; Becke, 1993; Hertwig & Koch, 1995). Optimized geometries of large models of intermediates 3R, 3I1 and 1I1 were obtained from the Supporting information of Hernández-Ortega et al. (2015). Solution energies (Tomasi & Persico, 1994; Mennucci & Tomasi, 1997; Cossi et al., 1998) in water (ε = 78.34) and chlorobenzene (ε = 5.7, mimicking the less polar environment of the protein active site) of the singlet and triplet states of these molecules were computed using the 6-31G(d,p) basis set complemented with diffuse functions on the oxygen atoms to allow a better description of the oxygen-based anionic species (henceforth referred to as basis set BS1).

Due to computational limitations, the search of minimum energy crossing points (MECP) between non-interacting singlet and triplet states required extensive trimming of the reaction model, which was reduced to the substrate, a water molecule, and the sidechains of the active-site dyad His251/Asp126 responsible for substrate deprotonation. MECP were located employing the methodology developed by Harvey et al. (1998) at the B3LYP/BS1 level. Since MECP optimization in gas phase yield very different geometries from continuum MECP optimizations (Silva & Ramos, 2007), we performed this search with a PCM continuum model using water as the solvent. The Cβ atoms of His251 and Asp126 were kept frozen to limit system flexibility to that possible in the enzyme active site. Investigation of CO release step were performed with the larger model suggested by Aitor-Hernández et al. (including the sidechains of His38, His100, Ser101, His102, Asp126, Trp160, His251, and the backbone amide linking Trp36 to Cys37), with several atoms kept fixed to prevent unrealistic movements. The fixed atoms were: Cβ of His100 and His102, Cα of Ser101, Trp36 and Cys37, Cβ and Cγ of His38, His251 and Trp160 and Cα and Cβ of Asp126. Very fine two-dimensional scans of the potential energy surface at the B3LYP/6-31G(d) level were performed by simultaneously varying the C3–C4 and the O–O distances (while keeping the C2–C3 distance fixed to prevent hysteresis). While the size of the system prevented the numerical computation (and updating) of the hessians needed for saddle point optimization, this scanning procedure allowed the generation of smooth potential energy surfaces which enabled the location of high quality transition structure guesses.

The activation energy of the one-electron transfer between substrates and O2 were estimated by applying Marcus theory for electron transfer, as suggested by Blomberg & Siegbahn (2003) and subsequently modified by Silva & Ramos (2008). As in previous works by our and other groups (Silva & Ramos, 2008; Silva & Ramos, 2009; Silva, 2014; Wijaya et al., 2016) reorganization energies for every molecule in both oxidation states were computed using the water-optimized reactant geometries for the product state (and vice-versa) and activation energies were then computed by building appropriate Marcus parabolas using these reorganization energies. The smaller size of these models allowed us to increase the size of the basis set in these computations to 6-311G(d,p), while keeping the diffuse functions on the oxygen atoms to allow a better description of the oxygen-based anionic species (henceforth this basis set will be referred to as BS2). Atomic charge and spin density distributions were calculated with a Mulliken population analysis (Mulliken, 1955) based on symmetrically orthogonalized orbitals (Löwdin, 1970).

Computation of the binding modes of 2-methyl- and 2-butyl-(1H)-3-hydroxy-4-oxoquinoline towards 2,4-dioxygenase (PDB:2WJ4 (Steiner et al., 2010)) were performed in YASARA Structure (Krieger & Vriend, 2014) using its AutoDock VINA module with default parameters (Trott & Olson, 2010). The docking region was confined to a 39.8 × 34.8 × 34.8 Å box centered on residues Trp36, His38, His100, Ser101, His102, Asp126, Trp160, and His251. Residues Gly35, Trp36, His38, His100, Ser101, His102, Leu128, Phe136, Leu156, Trp160, Met177, Trp185, Ile192 and His251 were kept flexible during the docking procedure.

Results and Discussion

1-H-3-hydroxy-4-oxoquinaldine-2,4-dioxygenase

We started the search for minimum-energy crossing points between the triplet and singlet states of O2 and deprotonated (1H)-3-hydroxy-4-oxoquinolines from the reported structures of the 3I1 intermediate. To keep the computations tractable most of the surrounding aminoacids were excised, and only the Asp-His dyad responsible for the initial deprotonation of substrate (Steiner et al., 2010; Hernandez-Ortega et al., 2014) and charge stabilization of the 3I1/1I1 intermediates was kept. Table 1 shows that this truncation has very modest effects on the reaction energetics, and should therefore not introduce relevant errors.

Table 1 Comparison of the quality of the energies obtained with the truncated model (which includes only the substrate and the Asp/His dyad) vs. the energies obtained with the large model used by Hernandez-Ortega et al. (2015).

All energies are computed vs. the respective reactant state at the B3LYP/BS1 theory level in water. The large model includes the sidechains of His38, His100, Ser101, His102, Asp126, Trp160, His251, and the backbone amide linking Trp36 to Cys37. All coordinates were taken from the Supporting information of Hernandez-Ortega et al. (2015) and used without further optimization.

Quinoline substituent	Model used	3I1	1I1	
-F	Large model	5.5	−10.1	
-F	His251/Asp126+ substrate	5.3	−10.1	
- CH3	Large model	12.6	0.6	
- CH3	His251/Asp126+ substrate	13.5	1.2	
-(CH2)4CH3	Large model	19.2	4.8	
-(CH2)4CH3	His251/Asp126+ substrate	18.7	1.5	
-NO2	Large model	19.0	7.2	
-NO2	His251/Asp126+ substrate	24.0	10.6	

Figure 2 Optimized B3LYP/BS1 geometries of the minimum-energy crossing points of (1H)-3-hydroxy-4-oxoquinolines bearing pentyl (A), methyl (B), fluoro (C) and nitro (D) substituents.

Spins on the oxygen atoms are shown for the triplet state at each of these geometries.

As in previous work by Hernandez-Ortega et al. (2014), all computations were repeated for four different (1H)-3-hydroxy-4-oxoquinolines to ascertain the influence of different substituents (methyl, pentyl, fluor and nitro) in the reaction course. The minimum-energy crossing points found (Fig. 2) were dramatically different from the 3I1 intermediates postulated in the previous work, which contain short (1.499–1.502 Å) substrate-oxygen bonds and longer O–O bonds (1.38 Å) than observed for free superoxide (1.334 Å). The sole exception was found to be the fluoro-substituted substrate, where the MECP geometry presented a short (1.56 Å) C–O distance (Table 2) and where the spin distribution at the triplet state (Fig. 2) was the most different from the initial reactant state. In spite of the large change relative to the initial state, the MECP for this substrate proved to be the most energetically accessible of all the tested quinolines (Table 2). Geometry optimizations of the triplet state starting from these MECP geometries invariably yielded the triplet reactant state and optimizations of the singlet state starting from this same geometry invariably collapsed into 1I1 intermediates. In contrast to the earlier workers, optimization of 3I1 intermediates proved to be impossible with HODMe or HODpentyl due to spontaneous break of the postulated substrate-O, yielding the original triplet reactant state. For HODF and HODNO2 the 3I1 intermediates proved to be local minima in the potential energy surface, as found by Hernandez-Ortega et al., but the C–O bond in 3I1 HODF was found to be 0.1 Å longer than found by them. This behaviour was observed both in the Asp/His only model (6-31G(d,p)(+) with a PCM continuum) and in the full model (6-31G(d)(+) in gas phase). Surprisingly, gas-phase optimization of the full system with 6-31G (as used by Hernandez-Ortega et al.) did not afford stable 3I1 with any substrate. I cannot tell whether the difference in results is due to subtle differences in the basis sets exponents, DFT integration grids, the particular solvation model used, to their choice of freezing all atoms in the active site aminoacids (vs. only a few backbone atoms in the present work) or to the particular choice of basis sets: the basis set they used for geometry optimizations (6-31G) lacks diffuse or polarization functions and is therefore less flexible/accurate than the 6-31G(d,p)(+) basis set I chose. The transition states leading from separated triplet reactants to these 3I1 intermediates lie less than 0.2 kcal mol−1 above 3I1,as also observed by Hernández-Ortega et al. The lack of minimum energy crossing points in the region of the potential energy surface leading from 3I1 to 1I1geometries entails that the reaction will most likely proceed directly through the MECP and thence to 1I1 and that in the few cases where the 3I1 intermediates are stable their formation is an unproductive side-reaction in spite of their energetic accessibility (Table 2).

Table 2 Characterization of the minimum-energy crossing points between the singlet and triplet surfaces of oxygen:(1H)-3-hydroxy-4-oxoquinoline systems in the presence of the His251/Asp126 catalytic dyad, at the B3LYP/BS1 level in a water continuum.

The Cβ atoms of His251 and Asp126 were kept frozen to limit system flexibility to that possible in the enzyme active site. n.a: not applicable, as the 3I1 species for these substituents are not local minima in the potential energy surface and collapse into separated substrate and triplet O2. For the -NO2 substituted quinoline, the substrate–oxygen distance in 1I1 is quite long (2.07 Å), and this intermediate is more properly described as a superoxide:substrate radical pair.

Quinoline substituent	-(CH2)4CH3	-CH3	-F	-NO2	
C–O distance (Å) at the MECP	2.308	2.23	1.568	1.968	
O–O distance (Å) at the MECP	1.303	1.307	1.326	1.304	
MECP energy (kcal mol−1) vs. reactants	16.8	15.2	9.2	24.2	
1I1 energy (kcal mol−1) vs. reactants	11.1	9.0	−3.4	23.6	
3I1 energy (kcal mol−1) vs. reactants	n.a	n.a	6.9	25.4	

A preference for direct electron transfer to O2 instead of a pathway relying on minimum-energy crossing points between surfaces of different spin multiplicity has been postulated before (e.g., Massey, 1994) for the flavin:O2 system and confirmed by quantum chemical computations (Prabhakar et al., 2002). Such preference is not limited to flavins, and has also been confirmed computationally for the deprotonated pyrrole in the reaction catalyzed by the oxygen-dependent coproporphyrinogen oxidase (Silva & Ramos, 2008). For the present enzymatic system, Hernandéz-Ortega et al. found that electron transfer from substrate to O2 was endergonic by 8–11 kcal mol−1, and concluded from these data (and from the inability of obtaining a converged wavefunction for the substrate radical:superoxide pair) that such a pathway would be impossible for this reaction. However, current SCF algorithms are well-known to have difficulties when attempting to enforce specific electron distributions (such as radical pairs lying at higher energies than the non-radical pair alternative), and therefore encountering convergence difficulties may not point to an intrinsic instability of those pairs but simply be a consequence of numerical/algorithmic difficulties. We have therefore analyzed the thermodynamic and kinetic feasibility of direct electron transfer from substituted quinolines to O2 using a Marcus formalism (Table 3). This formalism relies on the separate computation of the energies of electron-donor and acceptor, but such reliance is not expected to significantly detract from the quality of the results, as optimization of the radical pairs at large separations (20–120 ångstrom, to facilitate numerical convergence of the SCF procedure) yields energies which are very similar (within 0.5 kcal mol−1 in the gas phase) to the sums of the individually optimized superoxide and substrate radical (https://dx.doi.org/10.6084/m9.figshare.3426062.v22). The reaction rate was found to be strongly correlated with the electron-donating capability of the quinoline substituent (Table 3). For electron-donating and weakly-withdrawing substituents the reaction rate can be extremely fast, regardless of the polarity of the solvent. Polar environments generally lower the activation energy of this electron-transfer, enabling it to occur at rates exceeding 0.1 s−1 even for such electron-withdrawing substituents as acetyl, nitroso or nitrile. Overall, Table 3 shows that direct electron transfer from deprotonated co-substrate to O2 is predicted to be fast and uncomplicated even when the co-substrate bears moderately deactivating substituents. Comparison of these activation energies to the energies of the minimum-energy crossing points (Table 2) shows that for HODMe and HODpentyl the direct electron transfer route is favored over the reaction paths containing MECP (or the 3I1state found by Hernández-Ortega) by more than 8 kcal mol−1. The generation of the peroxide intermediate 1I1 is therefore most likely to proceed (in agreement with the proposal by Thierbach et al. (2014) and in contrast to the mechanism postulated by (Hernández-Ortega et al. (2015)) through electron transfer from substrate to O2, followed by recombination of the substrate-based radical with superoxide.

Table 3 Reaction energies and activation energies of the electron-transfer from substituted (1H)-3-hydroxy-4-oxoquinolines to dioxygen, at the B3LYP/BS2//B3LYP/BS1 level, computed using Marcus theory for electron transfer.

Unless otherwise noted, the 3-hydroxyl group remained in the deprotonated state. Substituents are shown ordered by increased values of their Hammet σm parameters (Hansch, Leo & Taft, 1991).

	In chlorobenzene	In water	
Quinoline substituent	Activation energy (kcal mol−1)	Reaction energy (kcal mol−1)	Activation energy (kcal mol−1)	Reaction energy (kcal mol−1)	
-NH2	3.4	−1.5	2.0	−4.8	
-COO−	1.5	−5.1	5.9	4.4	
-(CH2)4CH3	7.1	6.0	5.2	3.0	
-CH3	6.8	5.5	4.9	2.5	
-CH3 (protonated quinoline)	86.2	58.1	46.9	38.7	
-F	11.1	10.9	8.6	7.8	
-COCH3	22.5	21.5	16.6	16.5	
-CN	24.6	22.9	18.3	18.0	
-NO	23.9	23.7	20.0	20.0	
-NO2	41.2	33.1	31.1	27.5	

The endergonicity of the electron transfer step does not affect the viability of the process, since all it takes for a reaction to occur is that those endoenergetic steps are followed by sufficiently fast and exoenergetic steps so that the total activation barrier lies below ca. 20 kcal mol−1 (i.e., reaction rates above 10−2 s−1) and the overall reaction has a negative ΔG at the reactants/products concentrations present in the cell (so that the reaction proceeds in the direct rather than the reverse direction). In fact, superoxide formation has been experimentally confirmed in flavocytochrome b2 (Boubacar et al., 2007), which contains a FMN with a redox potential of −45 mV (Capeillère-Blandin, Barber & Bray, 1986) and from proline dehidrogenase (White et al., 2007), which harbors a FAD with a −75 mV redox potential, in spite of the endergonicity of the electron transfer from these cofactors to O2 (EmO2/O2 −  = −330 mV).

Hernández-Ortega et al. have shown that the peroxide intermediate 1I1 quickly becomes an endoperoxide (1I2) through attack of the substrate C4 by the terminal oxygen. Release of C=O from 1I2yields a carboxylate function on C4 and occurs quickly due to the stabilization of the nascent negative charge by hydrogen bonding with Ser101 (1.51 Å) and strong interaction with the positively-charged His251 (2.16 Å). In their computational investigation of this reaction step with quinolines bearing the much longer pentyl substituent, these researchers observed a remarkable increase of the activation energy for CO release of almost 16 kcal mol−1. Inspection of the structure of the transition state of the transformation of 1I2 into products reveals that the high activation energy of the pentyl-substituted quinoline is due to the use of the same binding mode for this quinoline as for the methyl-substituted quinoline, which introduces steric clashes between the pentyl-group and His38, His100 and the Trp36-Cys37 backbone. To avoid these clashes, the pentyl-substituted substrate is forced to rotate 30°around the axis perpendicular to the quinoline ring, thus increasing the separation between the substrate and Ser101 (to 1.85 Å) and His251 (to 2.67 Å), and strongly decreasing the charge stabilization provided by these residues on the nascent carboxylate (Fig. 3).

Figure 3 Proposed geometries of the transition states for the 1I2 → product reaction step for the (A) methyl-, and (B) pentyl-substituted 4-oxoquinolines.

Coordinates taken from the Supporting information of Hernández-Ortega et al. (2015). Trp160 has been omitted from images for clarity.

Figure 4 Newly-derived potential energy surfaces (at the B3LYP/6-31G(d) theory level) of the 1I2 → product reaction step for the (A) butyl-, and (C) methyl-substituted 4-oxoquinolines.

Geometries of the transition states for the 1I2 → product reaction step for the (B) butyl-, and (D) methyl-substituted 4-oxoquinolines are shown, with the substrate and sidechains of Ser101, Asp126 and His251 highlighted. Trp160 has been omitted from the images for clarity.

Although the 16 kcal mol−1 increase of activation energy for the CO release step in the pentyl-substituted was regarded by the original researchers (Hernández-Ortega et al., 2015) as “in agreement with the drop in [experimental] rate constant” reported earlier in the same paper for the butyl-substituted substrate, the observed 30% increase in kcat and 10-fold decrease of kcat/KM are not consistent with the 12 orders of magnitude difference in kcat expected from such a difference in activation energy. Additional evidence against the mechanistic relevance of the proposed binding mode for the pentyl-substituted substrate comes from the superposition of the transition state model coordinates with the crystallographic structure of the enzyme: even after this 30°rotation, the proposed position of pentyl group lies on the space occupied by the Pro35-Gly36 stretch of the enzyme, which had been left out of the active site model. Long hydrocarbon substituents may, however, be accommodated if a binding mode rotated by 240°is assumed, which places the aliphatic chain in the entrance channel bordered by Leu128, Phe136, Leu165, Val159, Trp160, Gln221 and His251. This binding mode was confirmed as the best hit in docking computations using Autodock VINA. A subsequent two-dimensional scan of the coordinates involved in the 1I2 → product transition showed that in this binding mode a very low energy pathway for CO release is accessible through a transition structure stabilized through interactions with Ser101 and His251 (Fig. 4B and Table 4). An identical scan was performed for the methyl-substituted quinolone in the original orientation (Table 4, Figs. 4C and 4D). The small differences in activation energies between both 4-oxoquinolines are fully consistent with the lack of dramatic differences in the experimentally-measured kinetic parameters.

Table 4 Comparison of the transition states of the CO release step for methyl- and butyl-substituted quinolones.

	Butyl quinolinea	Methyl quinolinea	Methyl quinolineb	
C3–C2 distance (Å)	1.71	1.77	1.749	
O–O distance (Å)	1.99	2.09	2.055	
TS energy vs. 1I2 (kcal mol−1) in water	8.1	11.3	8.0	
TS energy vs. 1I2 (kcal mol−1) (ε = 5.7)	8.0	11.4	8.4	
Ser101—O4 distance (Å)	1.57	1.58	1.741	
His251—O distance (Å)	1.89	1.83	Not applicable	
Notes.

a Structure obtained from very fine 2D-scans, with an active site model including the sidechains of His38, His100, Ser101, His102, Asp126, Trp160, His251, and the backbone of Trp36.

b Structure obtained from a complete saddle-point optimization in a minimal model including only the substrate, a water molecule and a methanol molecule mimicking Ser101.

Energies were computed at the B3LYP/BS2 level and do not include zero-point vibrational effects.

Other oxygenases

The experimental observation of EPR radical signals in anaerobic urate oxidase preparations upon incubation with uric acid (Gabison et al., 2011) is thought to support a reaction mechanism where urate dianion (generated through deprotonation of uric acid at the enzyme active site) transfers an electron to aminoacid sidechains (Lys, Arg or His) and reaction with O2 occurs through electron transfer from these aminoacid radicals. Our DFT computations (Table 5) show that direct electron from the urate dianion to O2 has such a low activation energy that no electron transfer to an active site aminoacid needs to occur to enable catalysis, and no minimum-energy crossing point between the singlet and triplet surfaces needs to be reached. The radical observed anaerobically (which may be His-based) should therefore play no role in the catalytic mechanism.

Table 5 Reaction energies and activation energies of the electron-transfer from urate dianion to dioxygen or aminoacid sidechains, at the B3LYP/BS2//B3LYP/BS1 level.

	In chlorobenzene	In water	
Electron acceptor	Activation energy (kcal mol−1)	Reaction energy (kcal mol−1)	Activation energy (kcal mol−1)	Reaction energy (kcal mol−1)	
O2	0.8	−9.3	4.2	0.1	
His+	18.8	17.0	48.1	47.5	
Lys+	34.3	30.9	65.2	64.3	
Arg+	24.5	20.5	51.5	51.4	

Finally, we computed the activation energy for the electron transfer between vitamin K and O2. The value obtained (5.3 kcal mol−1 in chlorobenzene, 3.3 kcal mol−1 in water) is, again, inferior to the energy of the minimum-energy crossing point between the singlet and triplet surfaces (15.3 kcal mol−1 in water (Silva & Ramos, 2007)). It thus appears that for all cofactor-less oxidases studied computationally so far (urate oxidase, 1-H-3-hydroxy-4-oxoquinaldine-2,4-dioxygenase, coproporphyrinogen oxidase and vitamin K-dependent glutamate carboxylase) catalysis occurs through direct electron transfer from substrate to O2 followed by radical recombination, instead of minimum-energy crossing points without formal electron transfer. A novel nogalamycin monooxygenase (Machovina, Usselman & DuBois, 2016) has recently been shown experimentally to also follow this paradigm, which thus appears to be the most general strategy for the enzyme-catalyzed reaction of triplet O2 with conjugated organic substrates in the absence of metal cofactors.

Conclusions

The computations described in this paper show that the previously postulated triplet endoperoxide intermediate (3I1) is most unlikely to play a role in the reaction mechanism of bacterial ring-cleaving 2,4-dioxygenase, as the minimum energy crossing point between the singlet and triplet surfaces directly connects the reactants to the singlet endoperoxide intermediate (1I1), and no MECP connecting the putative 3I1 intermediate to 1I1 is present. Moreover, the computed activation energy for the direct electron transfer from substrate to O2 is lower than the MECP energy for substrates bearing electron-donating or weak electron-withdrawing groups at the 2- position, enabling the formation of separated radical pairs after the 3-hydroxy group in the substrate is suitably deprotonated by the His251/Asp126 dyad (Table 2). Similar separated radical pairs also appear as the most likely intermediates in all other cofactor-less oxygenases studied computationally so far. The computation of small activation energies (comparable to those measured experimentally) using small models which do not take into account the full complexity of the active site/protein environment nor interact directly with the attacking dioxygen molecule strongly suggests that the additional complexity of the protein environment is not needed to “enhance the reactivity” of O2 per se, but simply to ensure that the substrates approach the critical portion of the active site (the Asp/His dyad responsible for the initial deprotonation) in the proper orientation. Finally, 1-H-3-hydroxy-4-oxoquinaldine-2,4-dioxygenase reactivity towards substrates bearing long alkyl chains on the 2-position is not possible in the originally postulated binding mode: the enzyme instead relies on a different binding mode which enables catalysis of the CO release step by positioning the nascent negative charges in a suitably stabilizing environment and affords reaction rates similar to those obtained with the sterically unencumbered HODMe substrate.

Supplemental Information

Supplemental Information 1 Optimized geometries

Cartesian coordinates of optimized MECPs and intermediates.

Click here for additional data file.

Supplemental Information 2 Electronic energies

Electronic energies of intermediates in the 1-H-3-hydroxy-4-oxoquinaldine-2,4-dioxygenase reaction mechanism.

Click here for additional data file.

Supplemental Information 3 Computation of reorganization energies for ring cleaving dioxygenase

Click here for additional data file.

Supplemental Information 4 Summary of activation energies for electron transfer in ring cleaving dioxygenase

Click here for additional data file.

Supplemental Information 5 Computation of reorganization energies for urate oxidase

Click here for additional data file.

Supplemental Information 6 Computation of reorganization energies for vitamin K reaction towards O2

Click here for additional data file.

Supplemental Information 7 Radical pair energies

Click here for additional data file.

Supplemental Information 8 Radical pair energies (BSSE-corrected)

Cartesian coordinates of optimized MECPs and intermediates.

Click here for additional data file.

Supplemental Information 9 Spin distribution (in the triplet state) at the MECP and 3I1 geometries for HODF

Click here for additional data file.

Supplemental Information 10 Spin distribution (in the triplet state) at the MECP and 3I1 geometries for HODNO2

Click here for additional data file.

Supplemental Information 11 Spin distribution (in the triplet state) at the MECP geometry for HODMe

Click here for additional data file.

Supplemental Information 12 Spin distribution (in the triplet state) at the MECP geometry for HODpentyl

Click here for additional data file.

Additional Information and Declarations

Competing Interests

Author Contributions

Data Availability

Pedro J. Silva is an Academic Editor for PeerJ. The author does not have any other competing interests.

Pedro J. Silva conceived and designed the experiments, performed the experiments, analyzed the data, wrote the paper, prepared figures and/or tables.

The following information was supplied regarding data availability:

Silva, Pedro (2016): Refining the reaction mechanism of O2 towards its substrate in cofactor-free dioxygenases. Figshare. https://dx.doi.org/10.6084/m9.figshare.3426062.v22.

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
