# Peer review of "Refining the reaction mechanism of O2 towards its co-substrate in cofactor-free dioxygenases"

_PeerJ, doi:10.7717/peerj.2805_

## Round 0.1 · original submission · Major Revisions

Please respond in detail to the reviewer's comment. A positive decision will be contingent on a positive response by both reviewers

Reviewer 1 ·

Basic reporting

This article is generally very well written with sufficient introduction and background that is used to demonstrate how this work fits into the wider research area. This work is also self-contained and detailed. There is one major problem, however, when one considers the availability of the raw data. I was unable to find any tables of spin densities / population. I find this extremely surprising since they are refereed to in the text and are of the UP MOST important when investigating this spin-forbidden reaction mechanism. It leaves many questions unanswered, about for instance whether the 1I1 intermediate is a closed or open shell species. Also please replace all the instances of the generic "ring-cleaving 2,4-dioxygenase" with "1-H-3-hydroxy-4-oxoquinaldine-2,4-dioxygenase (HOD)". Finally, please change (line 139) "As in the work we criticize" to "As in previous work by Hernandez-Ortega et al."

Experimental design

This work does include some important and new finding, however, when using a different software package and especially a modified basis set it is important to run full optimisation for all local minima and transition state structures. The problem with this approach is clearly demonstrated in Table 1 1I1, where the the large wt (R=CH3) system is only 0.6 kcal/mol higher in energy than 3R. This compares to an energy of approximately 10 kcal/mol reported by Aitor-Hernandez et al. Since the apparent driving force for this step is so much lower one may expect that 3TS1 is lower to. Therefore, the characterisation of this transition state is crucial to any further analysis.

Validity of the findings

In lines 142-144 the authors say that a C-O bond length of between 1.4 and 1.5 A is extremely short for a substrate superoxo. This is incorrect an indeed the xyz files (provided) show that the 1I1 structures have C-O bond lengths below this value. It is not at all surprising that there are longer bond lengths in the MECP structures, since this crossing point would presumably happen somewhere along the oxygen transfer step, however, this doesn't make the prier statement true. The MECP structures are very interesting, however, I am unsure about the conclusion the authors draw from these results. Firstly, the MECP energies are extremely close to the 3TS1 energies calculated by the previous JACs paper (within 2 kcal/mol). This is within the margin or error for this theory, also again I can not stress enough how problematic it is not to have 3TS1 calculated with the same very small cluster and basis set. The question of the direct electron transfer is more interesting still the values that are reported here (for the oxygen reduction) are again very close to the values reported before by the prier mentioned article. However, the main issue in the JACs paper was that the authors were unable to find a stable 2SUB* 2O2*- species. It is unclear that such a species has been found here. In line 221 the author indicate that CO release in the prier mentioned paper was changed by up to 20 kcal/mol depending on the substrate's R-group. This is indeed inaccurate. The affect of these substituents was assessed to be between 2.3 -5.1 kcal/mol. The misunderstanding here seems to be the switch in rate-limiting step with the various substrates. Finally, there is too little information on reactivity of urate oxidase (just 1 page) this needs to be either substantially expanded or else removed.

Additional comments

I would respectfully request that you read the above comments carefully. I my view the single most crucial thing that needs to be done is to find 3TS1 from 3R using the same model and basis set that was used for the MECP.

Reviewer 2 ·

Basic reporting

The manuscript is generally well written but tends to be scientifically inaccurate.
Several modifications should be done:
The title should be rewritten: oxygen and 1-H-3-hydroxy-4-oxoquinaldine are the dioxygenase substrates. Without enzyme there is not reaction between oxygen and the 1-H-3-hydroxy-4-oxoquinaldine.
- Abstract: “bacterial ring cleaving 2,4 dioxygenase” is not a valid name for an enzyme. Instead is recommended to use its full name (1-H-3-hydroxy-4-oxoquinaldine-2,4-dioxygenase) rather than other names. This applies for lines: 36, 106, 114, 276 and 291.
- Line 37: “flavin-like reactivity” is scientifically inaccurate. Radical pair mechanism is the correct term here. In general, the comparison that is made with respect to flavin reactivity is not correct from a chemically or an enzymatically point of view. Expressions as “flavin-like fashion” of “flavin-like reactivity” should be removed. The author should know that each flavoprotein is exhibiting different reactivity against oxygen, and therefore, different mechanism.
- Lines 40-43: Same citation appears twice. One should be removed. The paragraph is confusing, consider revision.
Some background literature needs to be updated or changed:
- Lines 19-21: The introduction should be mainly focus on the oxygen reactivity and its activation during enzymatic reactions. Some references here about proteins binding or generating oxygen, and the role of their transition metals, is either not clear or irrelevant.
- The article from Massey 1994, is not only quite old is additionally a chemistry paper. To show that radical pairs are formed in flavin-oxygenases, some examples of these enzymes in which the radical pair has been detected or computationally calculated, should be cited here; specially considering that the author states that the mechanism employed by flavin oxygenases (or even oxygenases) is similar to the one followed by cofactor free oxygenases.
- Lines 51-60: It seems that here has been clearly a misunderstanding from the data reported by Hernandez-Ortega et al. First, the 8-11 kcal/mol were calculated from the equation 5 in which only the electron transfer is considered and all the reaction species were optimized (and their energies calculated) separately. Second, please read the original article (page 7479, right column) in which is clearly reported that those two potential energy surfaces were calculated.

Experimental design

The description of the computational methodology needs to be clearer and separated in different subsections (one for each research question that is addressed). For example, it seems that two different systems are used but it is not clear which methodology is applied for each.
Apparently, most optimized structures were taken from the Supplementary Information from Hernandez-Ortega et al. article. If a different basis set or different software package is used, is crucial to run a full optimization (or reaction scan) for each local minima or transition state structures calculated. In other words, it is impossible to know whether the MECP or 3TS1 (from Hernandez-Ortega et al.) are higher in energy.

Validity of the findings

The most interesting aspect for discussion is about if the electron transfer happens prior to the oxygen attack to create a superoxide/substrate radical pair or not. However, it seems that the author have not got a fully optimised superoxide and substrate radical structure which is crucial to validate his findings. This is the main reason why this can not be assessed (as stated for example in lines 186-189). This optimised structure (with a radical on both the oxygen and the substrate) should be provided. Whether the spin crossing happens during the oxygen binding (as reported in the present work) or just after (as reported before) makes little difference to the general enzyme mechanism.
Line 172-175: It is true that a direct electron transfer was postulated for flavins by Massey, 1994, but the direct electron transfer taking place in glucose oxidase (Prabhakar, 2002) is a spontaneous exothermic process. By contrast, in the present work most of the obtained values for the electron transfer are endothermic. In this sense, this citation is not supporting the author findings.
Table 3: It is not clear what this data is it adding to the discussion about the reactivity of the enzyme or its mechanism.
The section about “other oxygenases” is brief and incomplete. This section should be either removed or extended. In addition, no computational details are included in the methodology.
Finally, on the abstract is stated that “the active site does not affect the reactivity of oxygen directly…” but is not clear where this is coming from based on the computational results. This conclusion is not supported enough by the present work; and, in fact, might be wrong.

---

## Round 0.2 · Major Revisions

Dr. Silva,

The letter you are receiving is generic and cannot be edited by an academic editor. Here, I like to emphasize that any positive decision regarding publication will be contingent on a satisfactory response particularly to the reviewer (#1) who requested again major revisions.

Sincerely
Reinhard Schweitzer-Stenner

Reviewer 1 ·

Basic reporting

Again, generally this article is well written and presented. The addition of the intermediate spin densities are helpful, however, may prove a little problematic upon further analysis.

Experimental design

Unfortunately, the extra calculation asked for are not presented, however, the extra information provided does raise some new worrying questions. The issue of freely optimised 3I1 structures may not be totally necessary in all cases. If the author is saying that a geometry scan of the dioxygen attack on several of the triplet surfaces leads to an excited state with the geometry of the intermediate and the electronics of the reactant? I am happy to except that the 3TS1 transitions are very close to the 3I1 energies and that this intermediates are unstable. The problem come from the fact that the calculated MECPs are not substantially lower then these states. The author still needs to provide the values of the 3TS1/3I1 and 3MECP structures at the same level of theory and using the same model. It is still unclear which pathway is lower, however, the argument is also not as important as it may at first seem. Either dioxygen binds via a dominant triplet surface and forms an unstable 3I1 intermediate that decays into a closed shell singlet or a spin state crossing occurs to form an open shell 1I1, which can relax to the same closed shell singlet. Importantly, no such open shell singlet is even calculated (since the author now explains that only the closed shell 1I1 was evaluated). The final problem relates to the much more interesting prospect of the initial electron transfer to the molecular O2. Indeed I am grateful for the extra information provided by the author and it is of interest that the radical reactant complex has been evaluated to some extent. However, one may have expected a scan from the closed shell 1I1 backwards to explore the possibility of a diradical 1TS1. Obviously, one may expect this step to be barrierless. However, without calculating this you are left with numbers that look remarkably similar to the energy of the diradical wt species predicted by Hernandez et al. and not much more.

Validity of the findings

no comment

Additional comments

Unfortunately, I am unable to recommend publication until extra calculations are provided

Reviewer 2 ·

Basic reporting

No Comments

Experimental design

No Comments

Validity of the findings

I have noticed that most of reviewers previous comments were successfully addressed by the author, and accordingly, few changes were made in the main article/supplementary information. However, for improving the quality of the article and clarification of the findings, I would strongly recommend to take into account the following considerations:

1. As mentioned before by reviewer 1 and by me, one of the most interesting arguing points is the optimisation of the 3I1 species in contrast with previous findings. It should be clearly stated here that the differences in the methodology, used in both articles, is leading to different computational results. For this reason, I would suggest to add the following paragraph (or something similar) - written by the author on the letter to the reviewers- to the discussion:

... with a PCM continuum) and in the full model (6-31G(d)(+) in gas phase).Gas-phase optimization of the full system with 6-31G (as used by Hernandez-Ortega et al.) did not afford stable 3I1 with any substrate. I cannot tell whether the difference in results is due to subtle differences in the basis sets used, integration grids, or the particular solvation model used, to their choice of freezing all atoms in the active site aminoacids (vs. only a few backbone atoms in my case) or to the particular choice of basis sets: the basis set they used for geometry optimizations (6-31G) lacks diffuse or polarization functions and is therefore less flexible/accurate than the 6-31G(d,p)(+) basis set I chose. The lack of minimum energy….


2. In line with the above, for reader clarification and improvement of the discussion, I strongly recommend adding the following paragraph (taken from the letter to the reviewers) in the results/discussion (radical pair section):

In the original article Hernandez-Ortega et al. did show that the radical pair is an “excited state” (as its wavefunction/electron density collapsed to a different, lower energy, solution during SCF computation) but then concluded that such observation meant that it could not play a role in the reaction mechanism. Current SCF algorithms do indeed face trouble when attempting to enforce specific electron distributions (such as radical pairs lying at higher energies than the non-radical pair alternative), but such numerical/algorithmic difficulties cannot be used to argue against the existence/stability of those species.


3. Clarification of Table 3. Again, adding the following in the main text my help the reader to get a clear idea of the results presented:

Table 3 shows that direct electron transfer from deprotonated co-substrate to O2 is predicted to be fast and uncomplicated even when the co-substrate bears moderately deactivating substituents. Comparison to Table 2 also shows that this direct electron-transfer should be faster than the reaction path involving MECPs.

Additional comments

I have noticed that most of my previous comments were successfully addressed by the author, and accordingly, few changes/additions were made in the main article/supplementary information. However, for improving the quality of the article and clarification of the findings, I would strongly recommend to take into account the above considerations.

---

## Round 0.3 · Minor Revisions

It is my understanding that one of the reviewers still wants some major revisions. However, based on the now positive assessment of the other reviewer I am inclined to accept the paper, but I like to give you an opportunity to respond to the points of reviewer 1 in a way you find appropriate. Please detail you response in the letter that accompanies your submission.

Reviewer 1 ·

Basic reporting

"No Comments"

Experimental design

The extra information provided by the author is encouraging. Although expected, when the barrierless 1I1 is added to the characterisation of the singlet radical reactant I believe the case for an 8-10 kcal/mol barrier for the electron transfer pathway is indeed strong. Again, this is extreme close to the value estimated in the JACs paper, however, the characterisation of this is important and the analysis valid. I also agree that this is the most interesting aspect of this study, however, I must again ask for the 3TS1 calculation.

Validity of the findings

I believe that unfortunately there is still some confusion about what I am asking for. To begin with I am not so much worried about the substrates with substituted R-groups (since the main focus this study is on the wild type reactivity). Secondly, as I keep saying, the characterisation of the energy of the radical reactant and the MECP are important. Lastly, this obsession with a "stable" 3I1 is really strange. Since, looking at the previous paper it is clear that 3TS1 and 3I1 are almost the same. It is clear from both your and their research the this intermediate is not stable and that it is a closed shell singlet that is responsible for all the further reactivity. The energy of your Wt MECP is 15.2 kcal/mol and the only energy we have for the 3TS1 is around 17 kcal/mol. Therefore, you can not rule out the 3TS1 pathway without finding a larger gap in your energies. I do not want to labour this point too much, however, it is a bit frustrating because whether there is a spin state crossing event that happens early in the reaction (as you hypothesise) or immediately after O2 binds is not really that important.

Additional comments

I do not want to come across as too critical because I do believe that this is an important article that does deserve to be published. However, please either find a wild type 3TS1 barrier that is greater than 2 kcal/mol higher in energy then your MECP or change you conclusions to say that the direct electron transfer appears to be favoured, with an energy that is more than 8 kcal/mol below either the MECP (15.2) or the 3TS1 (17.0).

Reviewer 2 ·

Basic reporting

No Comments

Experimental design

No Comments

Validity of the findings

No Comments

Additional comments

No Comments

---

## Round 0.4 · accepted · Accept

I found your response to the reviewer's comments quite satisfactory. The paper is now ready for publication.